



# Development and evaluation of spectral nudging strategy for the simulation of summer precipitation over the Tibetan Plateau

Ziyu Huang[1], Lei Zhong[1,2,3*], Yaoming Ma[4,5,6], Yunfei Fu[1]

[1]School of Earth and Space Sciences, University of Science and Technology of China, Hefei 230026, China
[2]CAS Center for Excellence in Comparative Planetology, Hefei 230026, China
[3]Jiangsu Collaborative Innovation Center for Climate Change, Nanjing 210023, China
[4]Key Laboratory of Tibetan Environment Changes and Land Surface Processes, Institute of Tibetan Plateau Research, Chinese Academy of Science, Beijing 100101, China;
[5]CAS Center for Excellence in Tibetan Plateau Earth Sciences, Beijing 100101, China
[6]College of Earth and Planetary Sciences, University of Chinese Academy of Science, Beijing 100049, China

*Correspondence to*: zhonglei@ustc.edu.cn (L.Z.)

**Abstract.** Precipitation is the key component determining the water budget and climate change of the Tibetan Plateau (TP) under a warming climate. This high-latitude region is regarded as "the Third Pole" of the Earth and the "Asian Water Tower" and influences the eco-economy of downstream regions. However, the intensity and diurnal cycle of precipitation are inadequately depicted by current reanalysis products and regional climate models (RCMs). Spectral nudging is an effective dynamical downscaling method used to improve precipitation simulations of RCMs by preventing simulated fields from drifting away from large-scale reference fields, but the most effective manner of applying spectral nudging over the TP is unclear. In this paper, the effects of spectral nudging parameters (e.g., nudging variables, strengths and levels) on summer precipitation simulations and associated meteorological variables were evaluated over the TP. The results show that using a conventional continuous integration method with a single initialization is likely to result in the overforecasting of precipitation events and the overforecasting of horizontal wind speeds over the TP. In particular, model simulations show clear improvements in their representations of downscaled precipitation intensity and its diurnal variations, atmospheric temperature and water vapor when spectral nudging is applied towards the horizontal wind and geopotential height rather than towards the potential temperature and water vapor mixing ratio. This altering to the spectral nudging method not only reduces the wet bias of water vapor in the lower troposphere of the ERA-Interim reanalysis ( when it is used as the reference fields) but also alleviates the cold bias of atmospheric temperatures in the upper troposphere, while maintaining the accuracy of horizontal wind features for the simulated fields. The conclusions of this study imply how reference fields errors impact model simulations, and these results may improve the reliability of RCM results used to study the long-term regional climate change.



## 1 Introduction

The Tibetan Plateau (TP) is referred to the Third Pole of the Earth and its average elevation is greater than 4,000 m. The TP has a complex topography and is also represented as the "Asian Water Tower". Its

powerful thermodynamic and dynamic effects not only significantly influence regional climate patterns and climate change but also have great impacts on the processes of the Asian monsoon system and westerlies (Immerzeel and Bierkens, 2010;Wu et al., 2007). Regional variations of atmospheric heating resources over the TP significantly regulate the summer Asian monsoon and associated precipitation (Zhao and Chen, 2001). Precipitation is an essential meteorological variable to reveal the water cycle

and surface energy balance (Palmer and Raisanen, 2002). It is thus imperative to realistically simulate variations of precipitation in both spatial and temporal distribution over the TP, which is also a vital component that can assess the performance of models (Bohner and Lehmkuhl, 2005;Karmacharya et al., 2017).

The major limitations for climate studies over the TP are attributed to the sparse observation datasets in

both spatial and temporal coverage. Consequently, atmospheric reanalysis products or global climate model (GCM) output are often used, but they are still too coarse to represent the complex terrain of the TP (Lang and Barros, 2004;Palazzi et al., 2013). In addition, the seasonal-to-interannual prediction of precipitation, especially with respect to extreme precipitation events over the southern and southeastern TP, remains poorly modeled by the GCM (Su et al., 2013;Wang et al., 2016). Regional climate models

(RCMs) with higher resolution is thus used to dynamically downscale the coarse output of coupled GCMs or reanalysis datasets (Glisan et al., 2013;Lo et al., 2008;von Storch et al., 2000). RCMs have been widely applied to study monsoon variations, regional climate change and perform well in precipitation predictions (Gao et al., 2015;Jiang et al., 2019). Compared to global reanalysis, RCMs generally produce better annual variation and long-term trends of precipitation over the TP during wet seasons (Gao et al.,

2015;Jiang et al., 2016).

However, these advantages of RCMs are limited to the time range of simulations. When forecast time is beyond 36 h, models' skill is gradually diminishing (White et al., 1999). This is mainly affected by the continuously accumulated deviations between the RCM's circulation fields and the large-scale driving fields as time progresses (Miguez-Macho et al., 2004;Waldron et al., 1996). As a solution, the nudging

method was proposed and applied for RCMs to ensure the simulated field be consistent with the large-scale driving fields while eliminating the spurious reflections of large-scale circulations inside the domain (Miguez-Macho et al., 2004;von Storch et al., 2000;Miguez-Macho et al., 2005).

Two forms of nudging technique, including grid nudging and spectral nudging, could be used in the Weather Research and Forecasting (WRF) model. Grid nudging in WRF adds one or more artificial

tendency terms to relax the model state toward the observed fields at each grid point, which is based on the differences between the model solution and the observations (Stauffer and Seaman, 1994). The horizontal wind, potential temperature and water vapor mixing ratio can be nudged in grid nudging, and it has been successfully used for regional climate simulations (Bowden et al., 2013;Bowden et al., 2012;Lo et al., 2008). However, grid nudging is spectrally indiscriminate, because it modulates the model

solution with the same strength, which neglects the features of the variations of interpolated small-scale for the model fields.



Meanwhile, spectral nudging is scale-discriminated, thus only the wavelength longer than the selected wavenumbers of model fields will be nudged towards the driving fields. Spectral nudging is capable to alleviate the bias while keeping the inner variability for model domain. Besides horizontal wind and

potential temperature can be nudged in spectral nudging, spectral nudging is also applied to geopotential. Compared with grid nudging, spectral nudging reduces suspicious reflections at the lateral boundaries and removes the influence of domain size and position of the RCM (Miguez-Macho et al., 2004). These advantages of spectral nudging not only improve the variability of mean or extreme precipitation simulations (Otte et al., 2012), but also efficiently improve large-scale atmospheric circulation

simulations (Bowden et al., 2012). By the efforts of Sepro et al. (2014), spectral nudging can be applied toward water vapor mixing ratio and restrict nudging toward potential temperature and water vapor mixing ratio above the tropopause. These modifications to fundamental spectral nudging strategy improve the 2-m temperature, upper-layer cloud cover and precipitation simulation over the North America and the contiguous United States (CONUS). Although various sensitivity experiments have

been devoted to examine the effect of spectral nudging on regional climate simulations in RCMs (Tang et al., 2018;Moon et al., 2018), studies on how to identify a most effective nudging strategy for precipitation simulation over the TP are limited. Because optimal nudging strategies of different regions may not be suitable for the TP with sophisticated and high topography. The effects of spectral nudging toward water vapor mixing ratio on precipitation simulation are also needed to be evaluated.

In this study, different spectral nudging strategies were conducted over the TP for the July of 2008. The influences of nudging parameters, including nudging variables, strength and model levels where nudging is applied, on precipitation simulations were explored, with a particular focus on the performance of extreme precipitation events. In the following sections, the WRF model, experimental design and validated data are described in section 2. Section 3 shows the validation of the various WRF simulations

against observation data, and analysis of the effects of spectral nudging on large-scale atmospheric circulation are discussed in section 4. In the section 5, the conclusions of the research are presented.

## 2 Model and experimental setting

### 2.1 WRF model configuration

The WRF model version 4.0 was used in this study. The capability of spectral nudging towards water

vapor mixing ratio was added in this version. A two-nested domain used in this study is displayed in Figure 1. The outer domain with 30-km spatial resolution provides the information from the large-scale processes. The inner domain with 10-km spatial resolution is the main study area and covers the complex terrain of the TP. The terrain of the region varies intensely in a short spatial range, which can exert strong turbulent drag on low-level atmospheric circulation and thus influence water vapor transport. To avoid

the influence of regional difference when calculating the mean precipitation over the entire TP, evaluations of precipitation simulations were also conducted on extreme (the highest 5 percentile value of) precipitation events.





### 2.2 Experimental settings

The physical options used in this study are followed to the settings of the High Asia Refined (HAR) data
(Maussion et al., 2011;Maussion et al., 2014). Specifically, the Thompson scheme (Thompson et al.,
2004), the Grell 3D ensemble scheme (Kain, 2004), the Dudhia shortwave radiation scheme (Dudhia,
1989) and Rapid Radiative Transfer model longwave radiation scheme (Mlawer et al., 1997), and the
unified Noah Land Surface Model (Chen and Dudhia, 2001) were selected, with the exception that the
Yonsei University scheme (Hong et al., 2006).

The initial and lateral boundary conditions were driven by the ERA-Interim reanalysis data with 6-h
temporal and 79 km spatial resolution (Dee et al., 2011). The sea surface temperatures were also derived
from ERA-Interim. All seven simulations, including one without nudging and six with spectral nudging
using various nudging strategies (Table 1), were initialized at 20 June 2008 and integrated continuously
for 40 days with first 10 days were set to spin-up time. The results of whole July of 2008 are used for
evaluation.

In spectral nudging simulations, cut-off wave number was set to a wavelength of 1000 km, for which has
been validated to be the most appropriate nudging wavelength to simulate precipitation events in many
regions (Gomez and Miguez-Macho, 2017;Yang et al., 2019). Therefore, the wavenumbers of $X$ and $Y$
directions were 5 and 4, respectively for the outer domain and 3 and 2, respectively for the inner domain.

Each nudging simulation was applied above the planetary boundary layer (PBL) so that allowing the
near-surface small-scale processes be freedom to respond to local processes.

The conventional continuous integration without nudging is designated "Control". Default spectral
nudging simulation is designated "SN". The nudging coefficient is related to the nudging relaxation time,
which indicates a predetermined time-scale how often nudging variables are relaxed toward the large-
scale driving fields. In SN simulation, nudging coefficients for horizontal wind, potential temperature,
and geopotential on both domains are the default value of $3.0\times10^{-4}$ s$^{-1}$ (relaxation time scale of 50 min).
For water vapor mixing ratio, default coefficient is $0.1\times10^{-4}$ s$^{-1}$ (relaxation time scale of 24 h). An
infinitely shorter relaxation time does not result a higher consistence between the simulated small-scale
fields with the forcing fields (Alexandru et al., 2009;Omrani et al., 2012). The relaxation time of nudging
variables is recommended to be equivalent to the temporal interval of the input driving data (Omrani et
al., 2013;Spero et al., 2018). Therefore, a nudging coefficient of $0.45\times10^{-4}$ s$^{-1}$ (relaxation time scale of 6
h) may be appropriate. Following this perspective, the subsequent sensitivity experiments were
conducted with a weaker nudging coefficient (longer relaxation time) for wind and temperature on basis
of SN, such as SNlowU and SNlowT simulations. Sensitivity simulation was not applied to geopotential,
because it has been shown a neglectable influence when spectral nudging toward geopotential with
different nudging coefficients (Spero et al., 2018). In WRF version 4, a new option "ktrop" was added to
allow spectral nudging be applied toward water vapor mixing ratio. This option adds a lid for both
potential temperature and water vapor mixing ratio fields at a predefined layer (nominally selected to
represent the tropopause) above the PBL, while horizontal wind component and geopotential are not
affected by the option.

As suggested from a 35-years analysis by using ERA-Interim reanalysis data, the pressure level of
tropopause over the TP varies between 93 and 106 hPa, with a mean value of 100 hPa during summer





(Zhou et al., 2019). In this study, the associated level of tropopause was 39 (namely, ktrop = 39). In this paper, it is important to examine the influence of adding the lid at different model layer, since the tropopause layer over the TP is much higher than other regions.

### 2.3 Validation data for model evaluation and comparison

Model simulated precipitation were evaluated against the merged Climate Prediction Center (CPC) MORPHing technique (CMORPH) precipitation dataset (hereinafter called CMORPH) with a temporal and spatial resolution of 1 h and 0.1°, which is developed by the China Meteorological Administration (CMA). The original CMORPH data with a spatial resolution of 8 km and temporal resolution of 30 min are firstly resampled to the 0.1° and 1 h spatial and temporal resolution. Then its systematic biases are corrected by 2400 rain gauges in China by using probability density function matching method. The corrected CMORPH is subsequently combined with hourly rain gauge analysis from automatic stations to derive the merged hourly precipitation dataset by applying the optimal interpolation methods (Joyce et al., 2004;Xie et al., 2017). Many studies demonstrate the high consistency and an acceptable bias of the CMORPH compared with the observed precipitation (Ou et al., 2020;Wei et al., 2018).

Beside the evaluation of precipitation, assessments of wind fields are also particularly important. The large-scale atmospheric circulation directly controls the atmospheric water vapor (AWV) transport between the TP and its surrounding area and exerts a great influence in the formation of precipitation. Impacts of different spectral nudging strategies on the 500-hPa AWV transport were compared with the fifth-generation global reanalysis of ECMWF, ERA5, which has a 31-km spatial resolution and 1-h temporal resolution. The horizontal wind fields of ERA5 over the TP have been verified with observations (He et al., 2019) and show the smallest wind bias compared to ERA-Interim reanalysis and an ensemble data assimilation system. In this study, ERA5 was interpolated to the 10-km downscaled grid resolution as the same to model output.

## 3 Results

### 3.1 Impacts of nudging strategy on mean precipitation simulations

The spatial consistency between simulated results and CMORPH was investigated by the monthly mean (larger than 0.1 mm/day) and extreme (the highest 5 percentile value of) precipitation (P95, mm/day; equals to 5.73 mm/day in this study). The monthly mean spatial distributions of the CMORPH precipitation fields of July and its difference with WRF simulations over the TP are depicted in Figure 2. As shown in Figure 2c, extreme precipitation can be observed over the southern and southeastern TP, and all simulations have large wet biases of precipitation compared to the CMORPH, especially along the southern edge of the Himalayas.

The overestimation of precipitation over the eastern TP was reduced when using spectral nudging method (Figure 2e), except that the use of spectral nudging with reduced nudging coefficient ($0.45\times10^{-4}$ s$^{-1}$) for potential temperature and horizontal wind. Both the SNlowU and SNlowT experiments showed little improvement in alleviating the wet bias of precipitation simulation (Figure 2f and 2g). Contrary to the expectations, SNQ_trop25 and SNQ_trop39 experiments ( adding nudging towards the water vapor



mixing ratio ) produce the largest wet bias of precipitation simulations, especially for extreme precipitation events over the southern TP (Figure 2i and 2j). Although SNnoT did not fully eliminate the over-forecasting of precipitation, it shown the smallest wet bias of extreme precipitation events over the southern TP (Figure 2h).

The spatial correlations were calculated from the time evolution of daily precipitation between CMORPH
and simulations at each grid cell over the TP (Figure 3). Overall, the correlation over the eastern TP in each experiment was found to be much higher than those over the western TP. Although model results have difficulties in capturing the temporal correlations with the CMORPH, some spectral nudging experiments significantly improved the correlation compared to Control simulation (Figure 3e, 3f and 3g), in which the correlation coefficients simulated by SNnoT and SNQ experiments were larger than
0.6 over the eastern TP.

The validations of root mean square error (RMSE) and mean absolute error (MAE) on the monthly mean and extreme precipitation (P95) events were also conducted with the CMORPH (Figure 4). Color shading represents the performance of each simulation, where more intense blue indicates a smaller bias of simulation and more intense orange indicates a larger bias of simulation.

Consistent with aforementioned results, the RMSEs of some spectral nudging simulations are not always superior to the conventional continuous integration simulation. Note that SNlowT and SNlowU produced errors comparable to the Control in terms of RMSE and MAE, whereas the default SN experiment performed worse than the Control experiments. Similar results were also found in the SNQ_trop25 and SNQ_trop39 experiments, where both experiments showed apparent higher RMSE and MAE compared
to Control, up to 23.10 mm day$^{-1}$ (RSME) in SNQ_trop39 for extreme precipitation events. The degraded skills of the SNQ_trop25 and SNQ_trop39 experiments in predicting precipitation were contrary to previous studies where adding spectral nudging toward water vapor mixing ratio could largely reduce the wet bias of precipitation intensity (Spero et al., 2014; 2018). In this assessment, the best performance was achieved by the SNnoT experiment with the lowest RMSE and MAE for different precipitation
thresholds. In addition, the SNnoT experiment showed a clear advantage for the extreme precipitation event, reducing the RMSE by 1.16 mm day$^{-1}$ compared with the Control and 2.27 mm day$^{-1}$ compared with the SN.

### 3.2 Evaluation of diurnal precipitation

In addition to error metrics, frequency distributions of precipitation intensity and diurnal cycle of the
mean precipitation from WRF simulations were compared with CMORPH (Figure 5). It is clear that precipitation from WRF simulations were heavily overestimated in the occurrence of precipitation events when precipitation intensity exceeds 3 mm day$^{-1}$. With respect to Control and other spectral nudging strategies, the frequency distribution of precipitation intensity simulated by the SNnoT experiment was more comparable to CMORPH (Figure 5a). The closer frequency density of high precipitation threshold
indicated the advantage of restricting nudging for temperature and water vapor mixing ratio in model may be attributed to decrease the overforecasting of extreme precipitation events over the southern TP.

The diurnal cycle of precipitation is an important feature of the monsoon precipitation because it controls the circulation characteristic and affects the precipitation magnitude (Bhatt et al., 2014;Sato et al., 2008).





To investigate whether using spectral nudging method can improve the simulation of diurnal cycle of precipitation, the variations of hourly precipitation from the CMORPH observations and model simulations are displayed in Figure 5b. According to the CMORPH observations, the averaged peaks of hourly precipitation mainly occurred in the afternoon and mid-night of the local time (UTC+8 h) and reached 0.2 mm h$^{-1}$. The precipitation forecast shows diurnal variation patterns comparable to those of CMORPH, but the simulated precipitation is remarkably larger during the day, which indicates that some individual precipitation events may be misrepresented. Compared with the Control, although the precipitation from SN showed improvement in terms of decreasing night-time precipitation overforecasts (the maximum of average precipitation was reduced by 0.05 mm/h), the precipitation simulation of the SN experiment performed worse during the day. The precipitation predictions from the SNQ_trop39 and SNQ_trop25 experiments, however, simulated a much higher wet bias, up to 0.1 mm/h, over the TP than did the Control. The smallest wet bias of precipitation was found in the SNnoT experiment during the day, which also outperformed other experiments at night, reducing the overforecast of precipitation by 0.03 mm/h compared with the Control and 0.05 mm/h compared with the SN. Consequently, the model simulations should be improved especially for precipitation forecasts during 2-12 UTC (10-20 of the local time), even though the precipitation forecasts of SNnoT greatly improved compared to the Control and SN forecasts. In addition, the precipitation forecasts of SNlowU and SNlowT had diurnal variations and comparable precipitation intensities that were similar to Control. These findings indicate that the SNnoT experiment had considerable improvements in simulating diurnal precipitation over the TP in comparison to the Control and SN experiments.

## 4 Discussions

### 4.1 Large-scale atmospheric circulation anomalies

In summer, precipitation occurred in the TP is mainly controlled by two water vapor channels, one of which is the transported by strong southwesterly under the effect of the Indian summer monsoon. The second channel is transported by the south branch of the mid-latitude westerly. By the influences of both channels, a large amount of AWV from the Bay of Bengal could be transferred the TP (Xu et al., 2008;Xu et al., 2002;Zhou and Li, 2002).

The monthly mean AWV transport fields, averaged for the whole atmospheric layer, related to the simulated precipitation are displayed in Figure 6. As represented by ERA5 (Figure 6a), large-scale atmospheric circulation favours strong AWV transport into the southeastern TP, with an AWV transport centre occurring at the windward slope of the southwestern Himalayas. All simulations reproduced such AWV fields over the southern TP but with weaker magnitudes compared to ERA5, in which more AWV is transported to the interior of the TP. Note that not all spectral nudging experiments reduced the wet bias of water vapor transport. Both SNlowU and SNlowT (Figure 6d and 6e) showed comparable results with those of the Control (Figure 6b), which indicated stronger AWV transport over the central and northern TP and overestimated the magnitude of the water vapor transport centre at the Bay of Bengal.

Despite the above three experiments (Figure 6b, 6d and 6e) reducing the AWV transport and causing smaller RMSEs in precipitation forecasts over the southern TP (Figure 4a), their AWV transport fields





were greatly misrepresented over the remaining regions of the TP, which could have detrimental effects on water cycle analysis. With reference to the above, the AWV transport obtained in SNnoT (Figure 6f) closely approached to that of ERA5, and the wet bias of AWV transport was reduced compared to the

other spectral nudging experiments and only indicated a slight overestimation over the western TP. As anticipated from the precipitation anomalies, such a difference in water vapor transport is obviously related to the smaller wet bias of precipitation.

The TP and its surrounding area have been regarded as a favorable region for convective processes because of its high elevation, by which moister can be transferred to the upper layers during summer

monsoon season (Fu et al., 2006; Heath and Fuelberg, 2014). Deep convection favors the process of transporting emissions from the surface into the upper-level atmosphere, through which moisture flux is vertically released into the atmospheric and influences the production of precipitation. SNQ_trop25 and SNQ_trop39 (Figure 6g and 6h) showed an analogous performance with smaller water vapor transport over the Bay of Bengal, indicating that the difference between SNQ_trop25 and SNQ_trop39 in

precipitation simulations may be attributed to local convection and subsequently processes.

### 4.2 Vertical structure of the convective process

The terrain of the southern Himalayas, which has a sharp altitude gradient, is extremely complex. Atmospheric water vapor transport is impeded, and a strong upward motion is consequently formed at the windward slope caused by the lifting effect of the complex terrain. However, the complex orography

of the Himalayas is greatly smoothed in RCMs, even though high horizontal spatial resolutions (e.g. 3 km) are applied (Wang et al., 2020). Therefore, the impact of a drag force due to the complex orography on the airflow is weakened, as is the convergence of water vapor. Consequently, more water vapor transport will be transferred to the high-latitude TP, causing more precipitation over the TP. Figure 7 and Figure 8 illustrate the monthly mean vertical structure of the specific humidity and vertical wind along

the averaged fields of 92-102°E and 28-34°N from seven experiments, respectively.

The strongest upward motion along the latitude was simulated by the SNQ_trop39 (Figure 7g) over the southern slope of the Himalayas, followed by the SN and SNQ_trop25 (Figure 7b and 7f). Large amounts of water vapor from the Bay of Bagel were transferred to the upper troposphere by the strong upward motion, which were converted to the interior of the TP through high-level southwestern advection,

subsequently causing more precipitation over those regions. Despite the SNlowU and SNlowT experiments (Figure 7c and 7d) simulating smaller upward wind compared to that of the SN, both patterns of the upward motion were similar with that of the Control (Figure 7a). The upward motion over the southern slope of the Himalayas simulated by the SNnoT (Figure 7e) showed a clear reduction compared to the other experiments, in which the upslope water vapor transport was largely limited by the

Himalayas. Therefore, most of the water vapor were condensed during upslope flow over the Himalayas, causing little water vapor available for precipitation over the interior of the TP.

Although the intensity of zonal mean upward motion over the TP is relatively weaker than that of meridional upward motion, a similar discrepancy was also observed in the SNnoT (Figure 8e), which indicated the weakest upward motion compared to the other experiments. Therefore, the convergence of

airflow accompanied by water vapor over the central TP was enhanced, leading to a slightly larger





precipitation intensity in the SNnoT (Figure 2h and Figure 6f) than in the other experiments. The strongest upward motion that occurred in the upper layer, along 88-102°E, was simulated by the Control (Figure 8a), followed by the SNQ_trop39, SN and SNQ_trop25 experiments. In the two SNQ simulations, the upward motion decreased when the lid was applied at a lower model level (layer 25, approximately 300 hPa in the model pressure level) (Figure 7f and Figure 8f). Large-scale driving datasets are insufficient to represent the horizontal and vertical variations in moisture and temperature above the tropopause (Miguez-Macho et al., 2004). The lid was originally designed to avoid the transferring moisture and temperature biases above the tropopause in the reference field to the model-simulated fields. With reference to the above analysis, it is suggested that setting the lid at the tropopause layer in the model is inappropriate when simulating precipitation over the TP.

### 4.3 Vertical profile of the atmospheric temperature, water vapor mixing ratio and horizontal wind speed

Dynamical downscaling studies in WRF simulations centred over the TP that use ERA-Interim reanalysis products as the reference fields have ensemble underestimations of atmospheric temperature and overestimations of the water vapor mixing ratio in summer (He et al., 2019; Xu et al., 2017). As the SNnoT experiment tends to exhibit smaller water vapor transport and weaker upward wind over the TP, the mean vertical profiles of atmospheric temperature and the water vapor mixing ratio from 100 to 700 hPa, averaged over the TP, were further investigated. The performances of the spectral nudging experiments were examined against the Control experiment and their differences are shown in Figure 9. Generally, apparent improvement is observed in several spectral nudging experiments in reducing the cold bias compared to the Control, except that the SN and SNQ_trop39 experiments resulted in larger cold biases in the upper troposphere, while the SNnoT and SNQ_trop25 experiments only had slightly smaller temperatures above 150 hPa (Figure 9a). Although the performances of SNlowU and SNlowT are highly comparable to the Control, both SNlowU and SNlowT simulated higher temperatures at lower levels. This pattern of SNlowU and SNlowT results is also observed in the comparison of the water vapor mixing ratios; both experiments simulated a slightly drier water vapor mixing ratio in the low troposphere (Figure 9b). The vertical profiles of the temperature difference fields of the SN, SNQ_trop25 and SNQ_trop39 showed similar variations below 300 hPa, but the temperatures in SN and SNQ_trop39 sharply decreased and became colder than those in the Control at higher layers. The vertical profile of SNQ_trop25, however, was aligned with the SNnoT in the middle troposphere but had higher temperature. In terms of the vertical spread of the water vapor mixing ratio (Figure 9b), the higher temperature obtained in SNQ_trop25 is likely attributed to its higher water vapor mixing ratio within the free troposphere, which caused wetter atmosphere conditions; thus, more heat flux was released once precipitation began.

Compared to the Control experiment, the best performance in reducing the wet bias of the water vapor mixing ratio in atmospheric layers was achieved by the SNnoT, despite the SN simulating slightly drier water vapor, with a value of 0.17 k kg$^{-1}$ at 400 hPa. Both SNQ_trop25 and SNQ_trop39 simulated the wettest water vapor mixing ratio at 500 hPa, but SNQ_trop25 had higher water vapor than did SNQ_trop39 when spectral nudging towards the water vapor mixing ratio was limited above 300 hPa in





the SNQ_trop25 experiment. Based on the above results, spectral nudging towards water vapor mixing ratio in more atmospheric layers will reduce the wet bias of water vapor. However, the improvement could not compensate for the negative impact that adding the ability to a model to perform spectral nudging towards water vapor artificially introduces extra wet bias due to the overestimation of ERA-Interim, when it is used as the reference field, in representing water vapor over the TP. In the study of He

et al., (2019), the evaluation of atmospheric temperature in ERA-Interim against observations showed an apparent cold bias in ERA-Interim above 400 hPa over the TP. The comparisons between SNnoT, SNQ_trop25 and SNQ_trop39 indicate that the cold bias of ERA-Interim in the upper troposphere had a higher contribution in to the deteriorating model's skill in simulating atmospheric temperatures than those of in the lower and middle troposphere. Therefore, to achieve an optimal strategy for precipitation

analysis and further water cycle budget studies without degrading temperature and water vapor simulations, model solutions must balance their accuracy in the lower and middle troposphere layers. In terms of the vertical profile of horizontal wind (Figure 9c), most spectral nudging experiments remarkably decreased the wind speed in the middle troposphere; however, the SNlowU and SNlowT showed consistent with that of Control.

In summary, the inter-comparisons of the ensemble model simulations demonstrate that SNnoT achieved notable improvements over the Control and the remaining spectral nudging experiments in water vapor transport, atmospheric temperature estimations and subsequent convective process simulations. The combinations of weakened convective motion over the Himalayan foothills and smaller horizontal wind speed in the free troposphere simulated by the SNnoT experiment limited water vapor to be transported

to the interior of the TP, which indicated less moisture available for precipitation. The improved moisture transport and large-scale circulation will have significant implications for precipitation analysis and water cycle budgets over the TP.

## 5 Conclusions

In this paper, the impacts and improvements of the spectral nudging technique in the WRF model for simulating precipitation and associated meteorological variables over the TP were evaluated using seven experiments.

Firstly, evaluations against the CMORPH precipitation data indicate that conventional continuous simulation cannot reproduce actual precipitation patterns and largely overestimate precipitation events

over the western and northern TP. The use of spectral nudging in the WRF model reduces such overestimations but does not always overcome all deficiencies when simulating the precipitation intensity and its diurnal cycle. Spectral nudging experiments with decreased nudging coefficients $(4.5 \times 10^{-5} \text{ s}^{-1})$ for wind and potential temperature showed comparable results with conventional continuous simulation in terms of precipitation, associated water vapor transport and temperature

simulations. In addition, allowing spectral nudging towards water vapor mixing ratio in more atmospheric layers within the troposphere can reduce the wet bias of water vapor. This improvement cannot compensate for the artificially introduced extra wet bias when water vapor in the model field is relaxed towards the ERA-Interim reanalysis. Therefore, although the ERA-Interim reanalysis has been





widely used as a large-scale reference field for regional climate studies of the TP in RCMs, its
uncertainties when representing atmospheric temperature and water vapor fields over this region should
be strongly considered when downscaling the ERA-Interim reanalysis. Following this perspective, the
evaluation of vertical temperature profiles implies that the simulations of atmospheric temperature in the
model are more sensitive to the cold bias of temperature due to ERA-Interim in the upper troposphere
than those in the lower and middle troposphere.

To decrease such biases and balance the accuracy of precipitation, temperature and water vapor
simulations, spectral nudging towards potential temperature and the water vapor mixing ratio was
restricted in the whole layers (designated SNnoT). The change to spectral nudging has clear advantages
over conventional continuous simulation and other spectral nudging experiments and largely improved
the precipitation intensity forecast as well as the forecast of its diurnal cycle compared to CMORPH.

Based on subsequent analysis, SNnoT reduced the meridional water vapor transport and upward motion
at the southern slope of the Himalayas; thus, less water vapor could reach the upper layers, which caused
less precipitation over the southern region and the interior of the TP. Consistently, SNnoT also improved
the simulations of atmospheric temperature and the water vapor mixing ratio by collectively alleviating
the cold bias of temperature and wet bias of the water vapor.

The evaluation and improvement of the spectral nudging technique in the WRF model in this work not
only concentrates on optimizing precipitation forecasts but also aims to increase the reliability of
RCM data used to assess the regional climate change without degrading the simulations of temperature
and water vapor. The conclusion of this work is also useful for the application of RCMs in dynamical
downscaling processes when using different reference fields. Since regional cumulus and land
surface processes are also essential in regional climate modelling because of their significant effects
on both large-scale and regional atmospheric circulation, improvements in representing such
physical processes are required in future studies.

**Data availability.** The ERA-Interim reanalysis data are available from the European Centre for Medium-
Range Weather Forecasts (ECMWF) Meteorological Archival and System (MARS) at
http://apps.ecmwf.int/datasets/data/interim-full-daily/levtype=sfc/ and
http://apps.ecmwf.int/datasets/data/interim-full-daily/levtype=pl/ (registration is required). The ERA5
data is available on the Copernicus Climate Change Service (C3S) Climate Data Store at
https://cds.climate.copernicus.eu/#!/search?text=ERA5&type=dataset/. The merged CMORPH
precipitation data that support the findings of this study are available from the corresponding author on
request.

**Author contribution.** Ziyu Huang conceived the initial idea of the work; Ziyu Huang, Lei Zhong,
Yaoming Ma and Yunfei Fu designed the method; Chun Zhao, Weiqiang Ma and Zhongbo Su helped
with the data processing; all authors helped in writing the paper.

**Competing interests.** The authors declare that they have no conflict of interest.





**Acknowledgements.** This research was jointly funded by the National Natural Science Foundation of
China (Grant No. 41875031), the Second Tibetan Plateau Scientific Expedition and Research (STEP)
Program (Grant No. 2019QZKK0103), the Strategic Priority Research Program of Chinese Academy of
Sciences (Grant No. XDA20060101), the Chinese Academy of Sciences Basic Frontier Science Research
Program from 0 to 1 Original Innovation Project (Grant No. ZDBS-LY-DQC005-01), the Chinese
Academy of Sciences (Grant No. QYZDJ-SSW-DQC019), the National Natural Science Foundation of
China (Grant No. 91837208, 41522501, 41275028), and CLIMATE-Pan-TPE (ID 58516) in the
framework of the ESA-MOST Dragon 5 Programme.

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



**Table 1: Nudging coefficients (10⁻⁴ s⁻¹) used for spectral simulations. Spectral nudging is indicated by "SN".**
***U, V* represent for horizontal wind component, *T* represents for potential temperature, *Q* represents for water vapor mixing ratio and *Φ* represents for geopotential. Nudging is applied to all layers above the PBL; layer 39 represents approximately the mean pressure level of tropopause over the TP; layer 25 represents approximately the normally lower limit for "ktrop" that is set to the middle layer in model simulation;**


| Simulation Name | Nudging Coefficients ($10^{-4}$ s$^{-1}$) | | | | |
|---|---|---|---|---|---|
| | *U,V* | *T* | *Q* | *Φ* | ktrop layer |
| Control | | | | | |
| SN | 3.0 | 3.0 | | 3.0 | |
| SNlowU | 0.45 | 3.0 | | 3.0 | |
| SNlowT | 3.0 | 0.45 | | 3.0 | |
| SNnoT | 3.0 | | | 3.0 | |
| SNQ_trop25 | 3.0 | 3.0 | 0.1 | 3.0 | 25 |
| SNQ_trop39 | 3.0 | 3.0 | 0.1 | 3.0 | 39 |



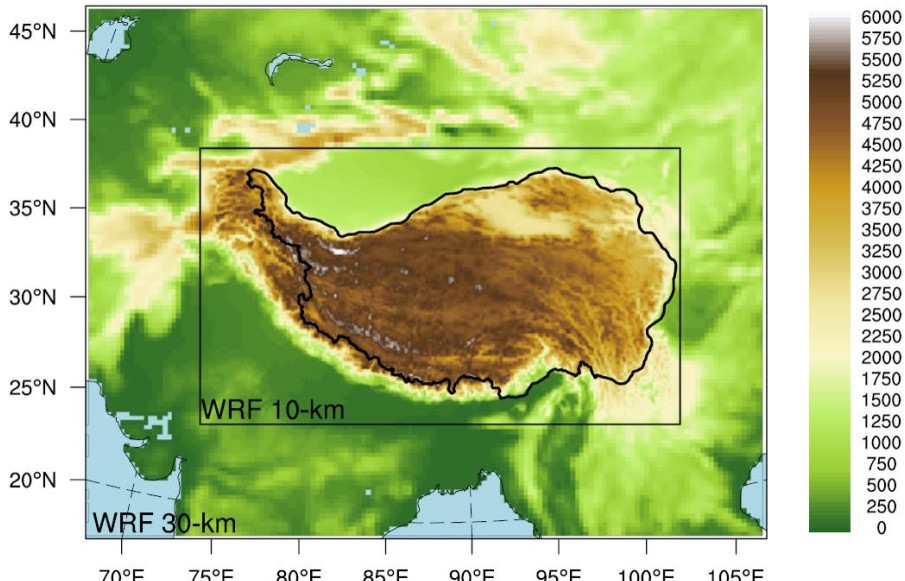

**Figure 1: WRF domains and model topography (unit: m). Black solid line in WRF 10-km domain represents the boundary of the TP.**






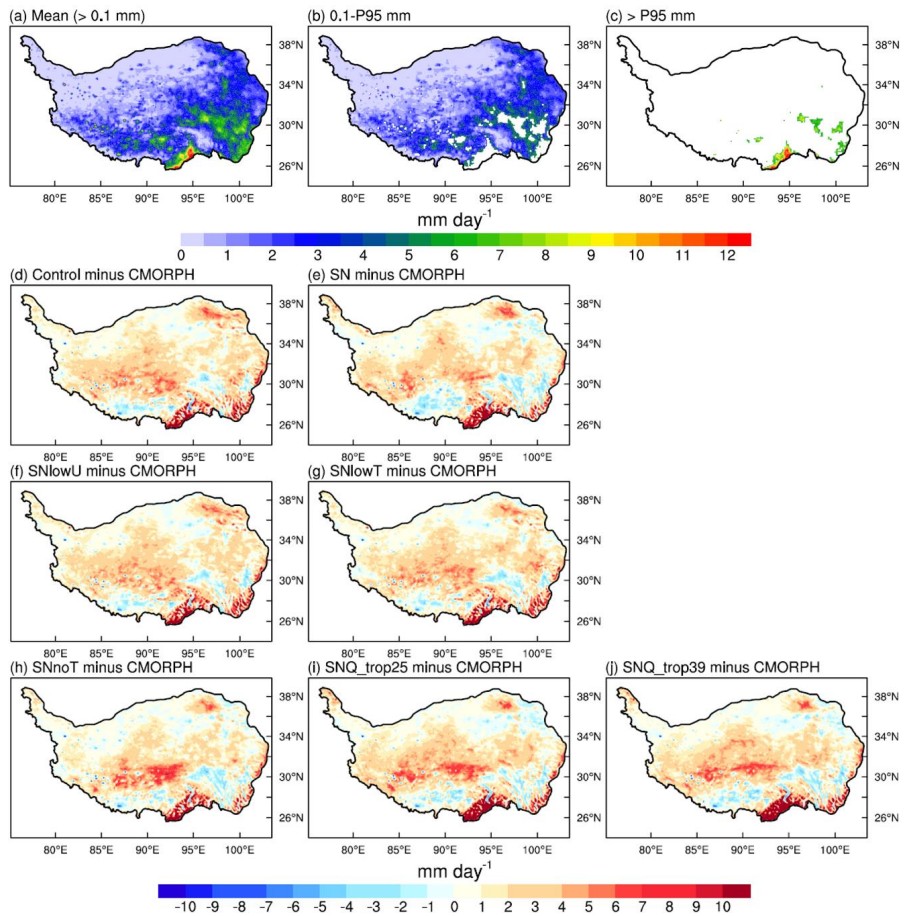

**Figure 2: Spatial distribution of mean and difference of precipitation (mm day⁻¹) fields for July 2008. (a) CMORPH, (b) CMOPH precipitation that smaller than P95 (5.73 mm day⁻¹) and (c) CMORPH precipitation where larger than P95. Differences between CMORPH and (d) Control, (e) SN, (f) SNlowU, (g) SNlowT, (h) SNnoT, (i) SNQ_trop25 and (j) SNQ_trop39, respectively. The bold black solid lines denote the boundary of the TP.**



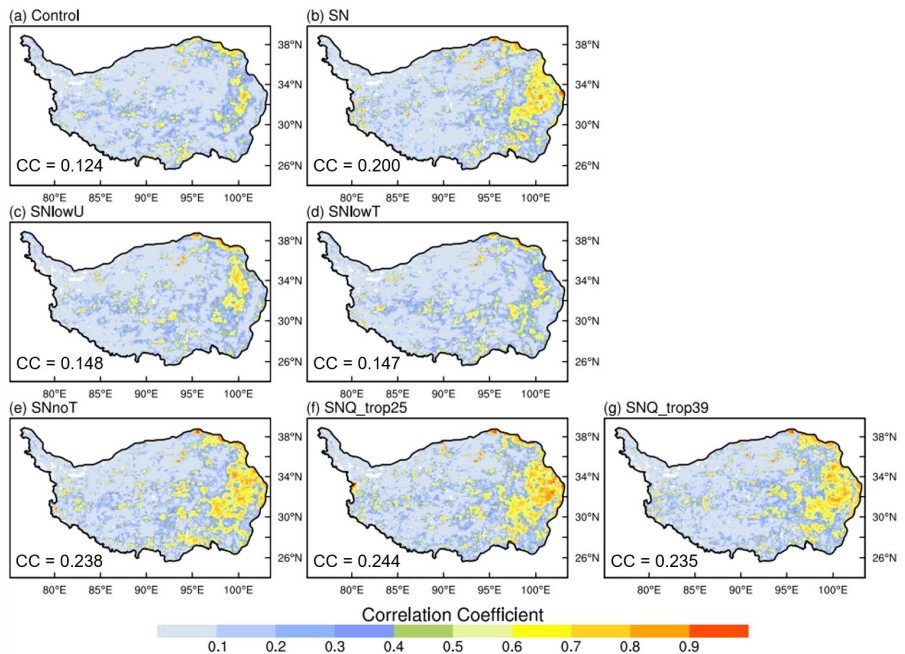

**Figure 3: Spatial distribution of the temporal correlation between CMORPH with (a) Control, (b) SN, (c) SNlowU, (d) SNlowT, (e) SNnoT, (f) SNQ_trop25 and (g) SNQ_trop39 over the TP.**





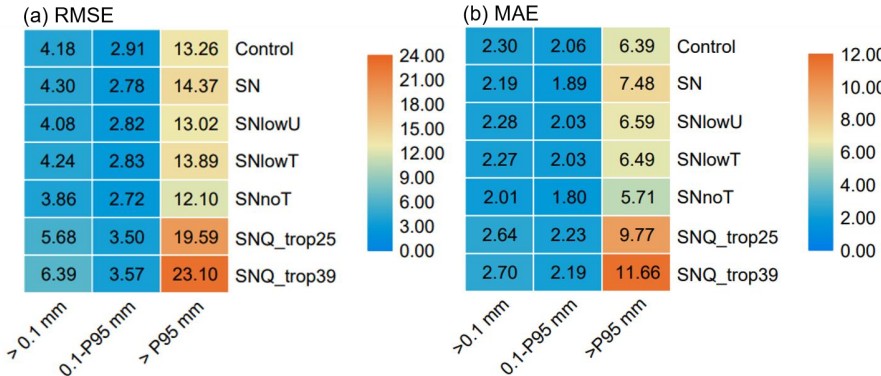

**Figure 4: Root mean square error (RMSE) and mean absolute error (MAE) of precipitation (mm day$^{-1}$) from WRF simulations against with CMORPH at different precipitation threshold (P95: monthly mean precipitation intensity at 5.73 mm day$^{-1}$). Color shading the performance of each WRF simulation, where more intense blue indicates a smaller bias of simulation and more intense orange indicates a larger bias of simulation.**





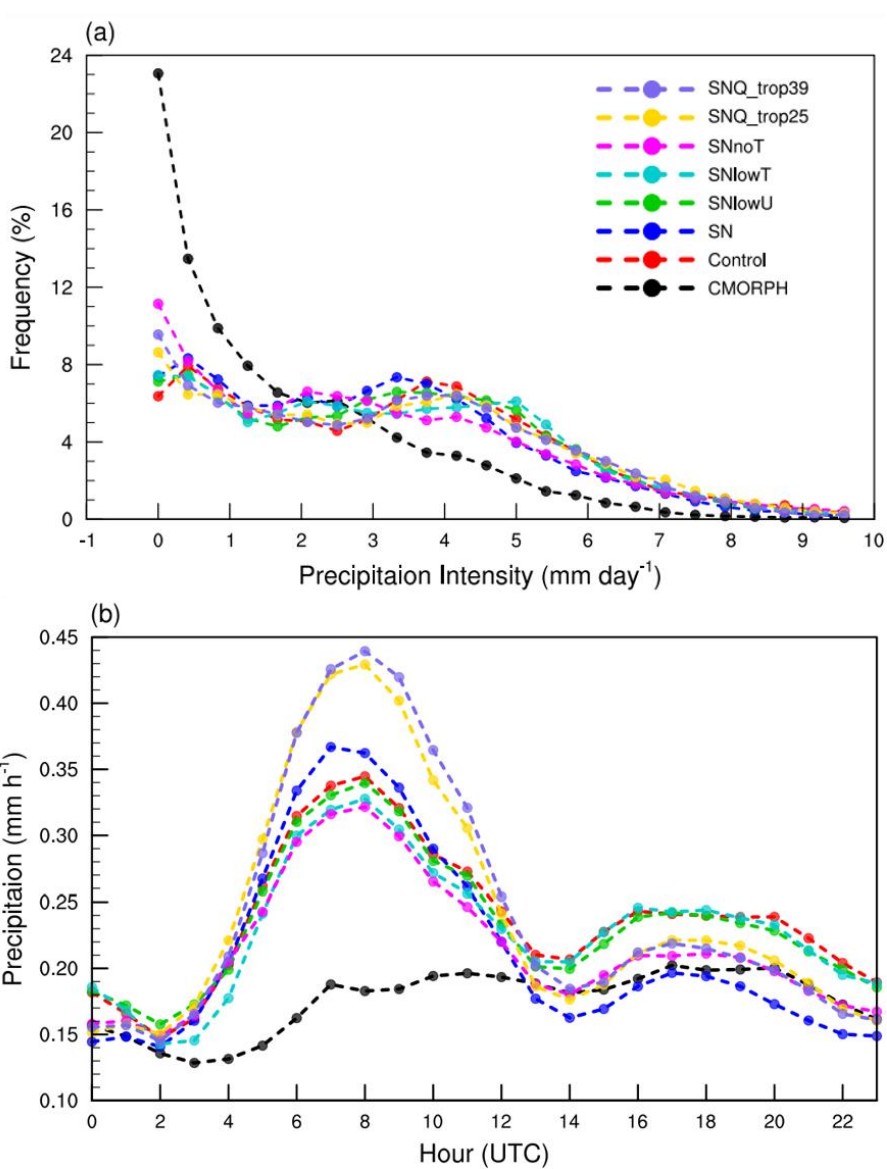

Figure 5: (a) the frequency distribution of mean precipitation intensity (mm day⁻¹) for CMORPH and simulated precipitation of July over the TP; (b) the average diurnal cycle of precipitation (mm h⁻¹) over the TP.



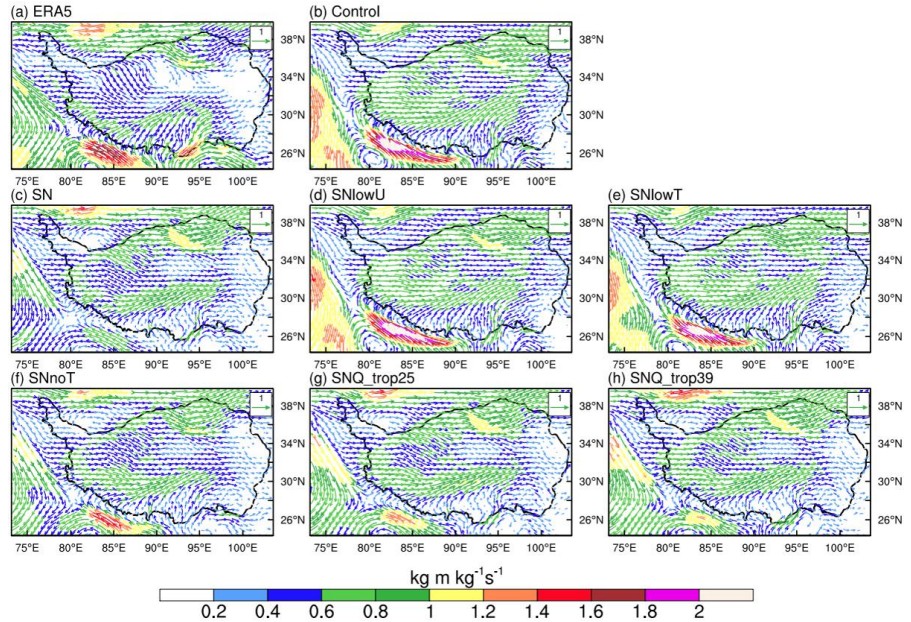

**Figure 6: Monthly mean atmospheric water vapor (AWV) transport fields (kg · m · kg$^{-1}$ · s$^{-1}$; horizontal wind multiples specific humidity and their values was magnified by 100 times) averaged in the whole atmospheric layers from (a) ERA5 and (b) Control, (c) SN, (d) SNlowU, (e) SNlowT, (f) SNnoT, (g) SNQ_trop25 and (h) SNQ_trop39.**



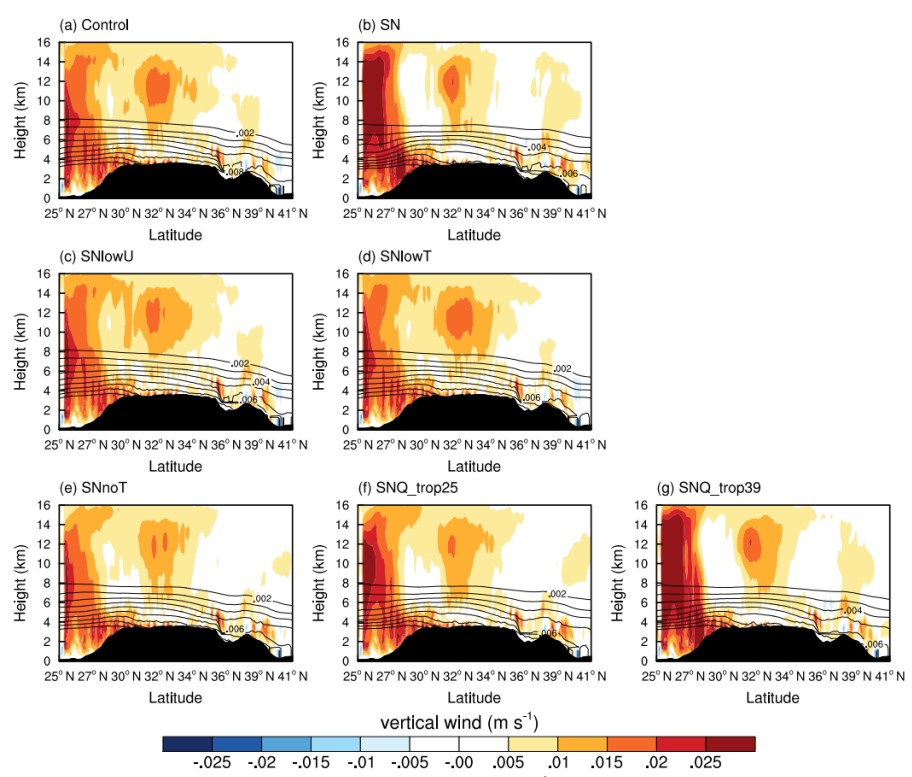

**Figure 7: Monthly mean vertical wind (color contour; m · s⁻¹; positive value means upward wind and negative value means downward wind) and specific humidity (contour line; kg · kg⁻¹) along the average of 92-102°E cross section for the simulations from (a) Control, (b) SN, (c) SNlowU, (d) SNlowT, (e) SNnoT, (f) SNQ_trop25 and (g) SNQ_trop39.**





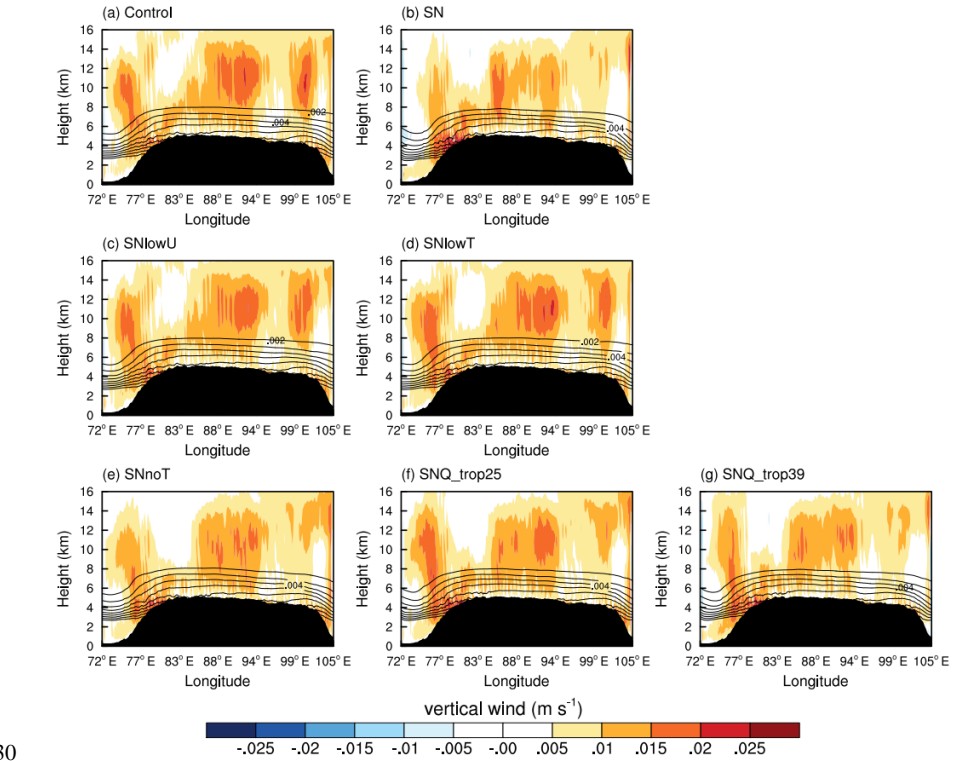

630

**Figure 8: Same as Figure 7, but along the average fields of 28-34°N.**



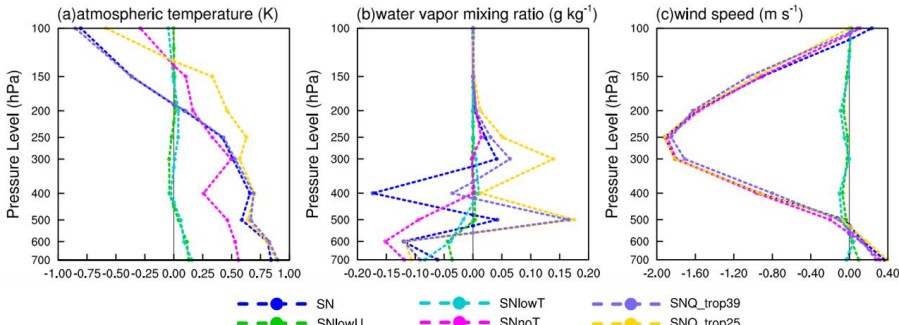

**Figure 9: Vertical profile of mean difference fields between Control and six spectral nudging experiments averaged over the TP for (a) atmospheric temperature (K), (b) water vapor mixing ratio (g kg⁻¹), (c) horizontal wind speed (m s⁻¹).**

635