# Peer review of "Development and evaluation of spectral nudging strategy for the simulation of summer precipitation over the Tibetan Plateau using WRF (v4.0)"

_Geoscientific Model Development, 2020_

## Short Comment (SC1) · 26 Feb 2021

Dear authors,

in my role as Executive editor of GMD, I would like to bring to your attention our Editorial version 1.2:

https://www.geosci-model-dev.net/12/2215/2019/

This highlights some requirements of papers published in GMD, which is also available on the GMD website in the 'Manuscript Types' section:

http://www.geoscientific-model-development.net/submission/manuscript_types.html

[Figure]

In particular, please note that for your paper, the following requirements have not been met in the Discussions paper:

- "The main paper must give the model name and version number (or other unique identifier) in the title."

- "If the model development relates to a single model then the model name and the version number must be included in the title of the paper. If the main intention of an article is to make a general (i.e. model independent) statement about the usefulness of a new development, but the usefulness is shown with the help of one specific model, the model name and version number must be stated in the title. The title could have a form such as, "Title outlining amazing generic advance: a case study with Model XXX (version Y)"."

Therefore please add a reference to the model (including version specifier) employed in you study, e.g. "Development and evaluation of spectral nudging strategy for the simulation of summer precipitation over the Tibetan Plateau using WRF (v..)"

Additionally, add a statement how to access the exact WRF code version you are using.

Yours,

Astrid Kerkweg

———————————————————

---

## Referee Comment (RC1) · Anonymous Referee #1 · 1 Mar 2021

A nicely written manuscript about downscaling of global reanalyses for regional climate modeling studies. I have no major comments, there are some spelling mistakes and incomplete sentences, but I am sure they will get corrected during the editing process.

I was missing a slightly more in-depth dynamical interpretation of why the best performing set of parameters improved the skill of WRF as an RCM.

Line 119-120: (incomplete sentence) Why did you use Yonsei scheme? I am assuming it is for the PBL.

Line 145: Is the 'ktrop' option for spectral nudging now available to the larger audience of WRF users?

Line 276: 'transporting emissions - which emissions?

Line 285: By 'drag force', do you mean orographic blocking? What is the convergence of the water vapor? Please clarify.

Line 315: It would be very instructive to overlay the 'ktrop' model level on figures 7 and 8. Is WRF using a terrain-following vertical coordinate? I imagine that the 'lid' is not horizontal at all - does that change your conclusions about how and where the nudging is applied?

Line 395: If I understand you correctly, the best results were achieved with SNnoT, nudging only the winds and geopotential, from the top of the PBL through the domain top, while not forcing the temperature and humidity at all?

---

## Referee Comment (RC2) · Anonymous Referee #2 · 2 Mar 2021

General comments: This paper evaluates the spectral nudging scheme implemented in the WRF model in order to evaluated the impact of spectral nudging on precipitation simulation over the Tibetan Plateau (TP). The spectral nudging is scale-discriminated so that only large spatial scales are constrained. The authors found that model simulations show clear improvements in their representations of downscaled precipitation intensity and its diurnal variations, atmospheric temperature and water vapor when spectral nudging is applied towards the horizontal wind and geopotential rather than towards the potential temperature and water vapor mixing ratio. The topic of the paper is interesting and it's a good fit for the scope of GMD. The description of the method is clear. The experiment design that testing nudging strength and nudging variables

is appealing and convincing. However, there are still some rooms for the improvement. For example, The evaluation of this study can be improved with some additional information. In the following, some comments are listed for author's revision.

1. The authors used the regional averaged difference between control (not nudged) simulation and several nudged simulations to show the impact of nudging on constrained fields. Please also quantify how well the nudged variables (e.g. T, U, V and Q) matched the ERAI reanalysis.

2. How the simulations of air temperature and water vapor mixing ratio are changed near the "lid" layer where the nudging is restricted toward both variables?

3. The large simulated precipitation bias and high precipitation RMSE over the Himalaya mountain region shown in Figure 2 and Figure 4 are possibly caused by the topographic differences between ERAI and WRF model. What are the impacts of nudging toward geopotential height on simulated results?

4. Please explain whether the nudging coefficient of horizontal winds and potential temperature shows similar results with control simulation.

5. Figure 7 and Figure 8 show the cross section of vertical wind and thus imply the convective processes over the TP and south slope of the Himalaya. I suggest that the authors should also consider the advection and atmospheric circulation in midtroposphere, which will make moisture lifted by convection to the interior of the TP.

6. Please add more details about the spectral nudging method.

7. Please use a more effective method to display the impacts of spectral nudging on specific humidity or atmospheric water content.

8. What is the accuracy of horizontal wind and atmospheric water transport fields in ERA5 over the TP?

---

## Author Comment (AC1) · 2 Mar 2021

Dear Astrid Kerkweg,

Thanks for your detailed comments. Our response to the short comments (SC) are given below.

SC: "The main paper must give the model name and version number (or other unique identifier) in the title." "If the model development relates to a single model then the model name and the version number must be included in the title of the paper. If the main intention of an article is to make a general (i.e. model independent) statement

[Figure]

about the usefulness of a new development, but the usefulness is shown with the help of one specific model, the model name and version number must be stated in the title. The title could have a form such as, "Title outlining amazing generic advance: a case study with Model XXX (version Y)"."

Response: The evaluation of spectral nudging strategy in this study was based on the simulations from WRF version 4.0. Therefore, according to the requirement, the title of this paper has been modified as: "Development and evaluation of spectral nudging strategy for the simulation of summer precipitation over the Tibetan Plateau using WRF (v4.0)".

SC: "Additionally, add a statement how to access the exact WRF code version you are using."

Response: The following statement has been added to the "Data Availability" in our revised paper. "The WRF v4.0 model code and documentation are freely available from the WRF website (https://www2.mmm.ucar.edu/wrf/users/download/get_sources.html)."

Best Regards,

Lei Zhong and Co-authors.

---

## Author Comment (AC2) · 9 Mar 2021

Response to Anonymous Referee #1

We would like to thank Referee #1 for the thorough review and constructive comments. We have carefully considered and addressed all comments, and significantly revised our manuscript. Please find our point-by-point response below. For clarity, the reviewer's comments are listed in black, while our response are shown in blue.

"A nicely written manuscript about downscaling of global reanalyses for regional climate modeling studies. I have no major comments, there are some spelling mistakes and incomplete sentences, but I am sure they will get corrected during the editing process."

**Author Response:** We would like to thank the reviewer for your positive comments on our study. We have taken each comment into account and revised our manuscript accordingly.

"I was missing a slightly more in-depth dynamical interpretation of why the best performing set of parameters improved the skill of WRF as an RCM."

**Author Response:** Thank you for this comment. To address this question, it is necessary to clarify the systematic inability of regional climate models first. Generally, a full forecast model within the regional domain can be decomposed into two components: the base field and the perturbation field. The base field is obtained from the driving fields, while the regional perturbation field is defined as the difference between the total field and the base field (Juang and Kanamitsu, 1994). Because of the systematic error of RCM, the inconsistencies between the model solution and the driving fields along the boundaries can cause large-scale errors (e.g., erroneous long waves), which might develop within the regional domain and produce undesirable noise.

In this case, spectral nudging technique is used to incorporate the large internal scales from outer model into the regional domain by adding a nudging term to the tendencies of nudged variables in model fields (Waldron et al., 1996). The nudging term is the summation of the difference between the spectrally expanded base field and the model field, multiplied by the nudging coefficients over selected wavenumbers. By applying spectral nudging technique, perturbation tendencies (a difference between the computed full field tendencies and the base field tendencies) are transformed to spectral space in RCM. Then, the spectral truncation filters out those waves longer than the regional domain from the perturbation tendencies. As a result, any scales longer than the regional domain cannot be modified during the course of integration (Kanamaru and Kanamitsu, 2007). Therefore, the large-scale information is retained and small-scale information can be freely resolved in downscaling process.

From the dynamical perspective, the best performing set of parameters of spectral nudging in this study is a combination of three corrections. For horizontal winds, spectral nudging determines the tendency of wind field perturbation whose scale was larger than the wavelength of 1000 km, so that the inconsistencies between the regional field and the base field can be alleviated. Nudging to geopotential height corrects the erroneous pressure difference due to the surface elevation difference between the ERA-Interim and WRF

model, leading to a better resolved topography of the Himalayas in the inner domain. For the temperature and humidity, their nudging coefficients were set to zero, which means their mean perturbations were set to zero, preventing the external forces on moisture and temperature (from nudging) to affect small-scale dynamics. In summary, nudging to winds and geopotential is strong enough to reduce large-scale errors in the regional domain, and this method improves the skill of WRF as a RCM.

Aforementioned dynamic interpretations of how spectral nudging influences the solution of RCM and how the set of spectral nudging parameters improved the skill of WRF have been added in 'Section 2.2' (Line 126-130) and 'Section 4.3' (Line 331-338) in the revised manuscript, respectively.

"Line 117-118: (incomplete sentence) Why did you use Yonsei scheme? I am assuming it is for the PBL."

**Author Response:** Thank you for this comment. A misleading sentence exists in the original manuscript. Actually, most physical options used in this study are the same as the options of the High Asia Refined (HAR) data. But for PBL scheme, we selected the YSU PBL scheme rather than Eta similarity PBL scheme. In the revised manuscript, relevant content has been revised as follows.
Line 118-119: 'The Yonsei University scheme (Hong et al., 2006) was used for the PBL scheme.'

"Line 145: Is the 'ktrop' option for spectral nudging now available to the larger audience of WRF users?"

**Author Response:** Thank you for this comment. To our knowledge, this new 'ktrop' option has been added in the released WRF model v4.0, and WRF users can use this option via 'namelist.input'. More details and usages about 'ktrop' are available from the source code 'WRF/phys/module_fdda_spnudging.F'                                  (https://github.com/wrf-model/WRF/blob/master/phys/module_fdda_spnudging.F#L266).

"Line 276: 'transporting emissions - which emissions?"

**Author Response:** Thank you for this comment. The 'emissions' here means water vapor. In the revised manuscript, relevant text now reads:
Line 304-307: 'Deep convection favors the process of transporting water vapor into the upper-level atmosphere, through which moisture flux is vertically released into the atmosphere and influences the formation of precipitation during summer monsoon season (Fu et al., 2006; Heath and Fuelberg, 2014).'

"Line 285: By 'drag force', do you mean orographic blocking? What is the convergence of the water vapor? Please clarify."

**Author Response:** Yes, 'drag force' here is relevant to the orographic blocking. Specifically,

it means the sub-grid orographic drag on water vapor due to the complex terrain over the southern slope of the Himalayas.

The convergence of water vapor is mainly caused by the barrier effect of the Himalayas, which are more than 5,000 m on average. In summer, abundant northward water vapor transport, originating from the surrounding oceans, impinges on the Himalayas. Upslope moisture transport is strictly limited by the sufficiently high orograph, and thus most of water vapor are 'wrung out' here, causing the convergence of water vapor (Dong et al., 2016).

To our knowledge, model with coarse spatial resolution ignores the impact of mesoscale and microscale orography. Accordingly, the surface friction and turbulent form drag on airflow with water vapor are reduced (Wang et al., 2020). Therefore, the convergence and condensation of water vapor over the southern slope of the Himalayas are weakened. In this case, more water vapor can be transported to the higher area and even the interior of the Tibetan Plateau.

Relevant text has been added to the revised manuscript as follows:

Line 268-277: 'The terrain of the Himalayas, featured by a sharp topography gradient, is more than 5,000 m on average and is regarded as a natural barrier for northward atmospheric flow. In summer, large amount of atmospheric water vapor transport, originating from the surrounding oceans, impinges on the Himalayas. The sufficient high topography strictly limits the upslope water vapor transport, and strong upward motions are consequently formed by the lifting effect of the complex terrain (Dong et al., 2016). However, the complex orography of the southern slope of the Himalayas is greatly smoothed in current RCMs. Accordingly, the surface friction and sub-grid orographic drag due to the impact of mesoscale and microscale orography on the airflow are weakened, which in turn reduce the convergence and condensation of water vapor over the southern slope of the Himalayas. Consequently, more water vapor transport could arrive the high-latitude TP, causing more precipitation over the TP.'

"Line 315: It would be very instructive to overlay the 'ktrop' model level on figures 7 and 8. Is WRF using a terrain-following vertical coordinate? I imagine that the 'lid' is not horizontal at all - does that change your conclusions about how and where the nudging is applied?"

**Author Response:** Thanks for your comments and suggestions. As shown in new Figure 7 and Figure 8, the 'ktrop' model level has been added in the revised manuscript and relevant explanation is available at Line: 308-311.

In this study, the WRF model used "Hybrid Vertical Coordinate". As explained by the WRF source code documentation 'README.hybrid_vert_coord' (https://github.com/NCAR/WRFV3/blob/master/README.hybrid_vert_coord), the hybrid vertical coordinate are terrain following near the surface, but then relax towards an isobaric surface aloft. The purpose of this coordinate option is to reduce the artificial influence of topography towards the top of the model.

Yes, you are correct. This lid is indeed not horizontal. Ideally, the user-defined lid (KTROP) is selected to represent the tropopause, and will be refined or supplemented with the space and time varying tropopause. The variability of the tropopause over the Tibetan Plateau (TP) is complicated during summer. In this study, it is an effective way to select a certain

pressure layer in the model vertical layers representing the tropopause as the lid. It does not change our conclusions about how and where the nudging is applied.

"Line 395: If I understand you correctly, the best results were achieved with SNnoT, nudging only the winds and geopotential, from the top of the PBL through the domain top, while not forcing the temperature and humidity at all?"

**Author Response:** Thank you for this comment. Yes, your understanding are correct. The best results in improving the simulation of the intensity and diurnal cycle of precipitation were achieved when spectral nudging was applied towards winds and geopotential and not forcing the temperature and humidity at all. The reasons can be summarized as follows: In this study, we have already run the sensitive experiments of nudging toward temperature and humidity on precipitation simulations, such as:

EXP1, 'SN', with nudging towards horizontal wind components (U, V), temperature (T) and geopotential height (G), and their nudging coefficients were all set to 0.0003 s$^{-1}$.

EXP2, 'SNlowT' with nudging towards U, V, T and G, but the nudging coefficient of T was 0.000045 s$^{-1}$;

EXP3, 'SNnoT' with nudging toward U, V and G only;

EXP3, 'SNQ_trop25' with nudging towards U, V, T, G and Q, while restricting nudging towards T and Q above the model layer of 25 (the lower limit for the 'lid' layer);

EXP4, 'SNQ_trop39' is same as 'SNQ_trop25' but restricting nudging towards T and Q above the model layer of 39 (approximate the tropopause layer).

A new Figure 6 has been given to evaluate above experiments for column-integrated water vapor transport. Compared to ERAI (Figure 6a1) and ERA5 (Figure 6b1), SN (Figure 6d1) misrepresented the large-scale northward water vapor transport. The impact of nudging towards temperature can be found in the comparison between SNlowT (Figure 6f1) and SNnoT (Figure 6g1), in which SNlowT simulated much stronger northward water vapor transport than SNnoT over the southeastern TP. This excessive water vapor transport can also be observed in SNQ_trop25 (Figure 6h1 and Figure 6h2) and SNQ_trop39 (Figure 6i1 and Figure 6i2).

A new Figure 7 has been given to evaluate the impact of nudging on convection over the southern slope of the Himalayas. From the Figure 7g, the strongest upward wind was simulated by SNQ_trop39, followed by SN (Figure 7b) and SNQ_trop25 (Figure 7f). In this case, large amounts of water vapor can be transported to the upper troposphere by strong upward motion, and then conveyed to the interior of the TP through upper-level atmospheric circulation. The upward motion over the southern slope of the Himalayas simulated by the SNnoT (Figure 7e) showed a clear reduction. Therefore, most of the water vapor were condensed in upslope flow over the Himalayas, causing less water vapor available for precipitation over the interior TP.

Accordingly, the smallest RMSE and MAE of precipitation simulation were achieved by SNnoT (Figure 4) rather than SNlowT, SNQ_trop25 or SNQ_trop39. In addition, larger RMSEs and MAEs were obtained when nudging towards humidity was applied. Therefore, spectral nudging was applied towards winds and geopotential only while not forcing the temperature and humidity at all.

Aforementioned new Figure 6 and Figure 7 have been added in the revised manuscript and relevant descriptions have also been added in 'Section 4.1' (Line: 278-301) and 'Section 4.2' (Line: 312-331), respectively.

[Figure]

Figure 6: Column-integrated northward water vapor transport (meridional wind component multiples by specific humidity, units: g m kg$^{-1}$ s$^{-1}$) averaged over the study period over the central Himalayas derived from (a1) ERAI (ERA-Interim), (b1) ERA5, (c1) Control, (d1) SN, (e1) SNlowU, (f1) SNlowT, (g1) SNnoT, (h1) SNQ_trop25 and (i1) SNQ_trop39, respectively. (a2)-(i2) are the same as (a1)-(i1) but for the eastward water vapor transport (zonal wind component multiples by specific humidity).

[Figure]

Figure 7: Vertical wind (m s⁻¹; positive value means upward wind and negative value means downward wind) averaged over the study period along the average of 92-102 °E cross section derived from (a) Control, (b) SN, (c) SNlowU, (d) SNlowT, (e) SNnoT, (f) SNQ_trop25 and (g) SNQ_trop39. Black solid lines represent the height of 'ktrop' layer of 25 and 39.

[Figure]

Figure 4: Root mean square error (RMSE) and mean absolute error (MAE) of precipitation (mm day⁻¹) from WRF simulations against CMORPH at different precipitation threshold (P95: monthly mean precipitation intensity at 5.73 mm day⁻¹). Color shading represents the performance of each WRF simulation, where more intense blue indicates a smaller bias of

simulation and more intense orange indicates a larger bias of simulation.

References:

Dong, W. H., Lin, Y. L., Wright, J. S., Ming, Y., Xie, Y. Y., Wang, B., Luo, Y., Huang, W. Y., Huang, J. B., Wang, L., Tian, L. D., Peng, Y. R., and Xu, F. H.: Summer rainfall over the southwestern Tibetan Plateau controlled by deep convection over the Indian subcontinent, Nat Commun, 7, 2016.

Hong, S. Y., Noh, Y., and Dudhia, J.: A new vertical diffusion package with an explicit treatment of entrainment processes, Mon Weather Rev, 134, 2318-2341, Doi 10.1175/Mwr3199.1, 2006.

Juang, H. M. H., and Kanamitsu, M.: The Nmc Nested Regional Spectral Model, Mon Weather Rev, 122, 3-26, 1994.

Kanamaru, H., and Kanamitsu, M.: Scale-selective bias correction in a downscaling of global analysis using a regional model, Mon Weather Rev, 135, 334-350, 2007.

Waldron, K. M., Paegle, J., and Horel, J. D.: Sensitivity of a spectrally filtered and nudged limited-area model to outer model options, Mon Weather Rev, 124, 529-547, 1996.

Wang, Y., Yang, K., Zhou, X., Chen, D. L., Lu, H., Ouyang, L., Chen, Y. Y., Lazhu, and Wang, B. B.: Synergy of orographic drag parameterization and high resolution greatly reduces biases of WRF-simulated precipitation in central Himalaya, Clim Dynam, 54, 1729-1740, 10.1007/s00382-019-05080-w, 2020.

---

## Author Comment (AC3) · 9 Mar 2021

**Response to Anonymous Referee #2**

We would like to thank Referee #2 for the insightful and constructive comments. All your comments and suggestions are very helpful for improving our manuscript. We have carefully considered and addressed all of these comments, and significantly revised our manuscript. Please find our point-by-point response below. For clarity, the reviewer's comments are listed in **black**, while our response are shown in blue.

"This paper evaluates the spectral nudging scheme implemented in the WRF model in order to evaluate the impact of spectral nudging on precipitation simulation over the Tibetan Plateau (TP). The spectral nudging is scale-discriminated so that only large spatial scales are constrained. The authors found that model simulations show clear improvements in their representations of downscaled precipitation intensity and its diurnal variations, atmospheric temperature and water vapor when spectral nudging is applied towards the horizontal wind and geopotential rather than towards the potential temperature and water vapor mixing ratio. The topic of the paper is interesting and it's a good fit for the scope of GMD. The description of the method is clear. The experiment design that testing nudging strength and nudging variables is appealing and convincing. However, there are still some rooms for the improvement. For example, the evaluation of this study can be improved with some additional information. In the following, some comments are listed for author's revision."

**Author Response:** Thank you for your positive comments and encouragement for our study! We have fully addressed every comment and revised our manuscript accordingly.**

"1. The authors used the regional averaged difference between control (not nudged) simulation and several nudged simulations to show the impact of nudging on constrained fields. Please also quantify how well the nudged variables (e.g. T, U, V and Q) matched the ERAI reanalysis."

**Author Response:** Done. Following your suggestion, vertical profile of zonal wind (U; m s-1), meridional wind (V; m s-1), atmospheric temperature (T; K) and specific humidity (Q; kg kg-1) of the difference between ERAI and model's simulation is shown in the new Figure 9. Generally, applying spectral nudging method in WRF model leads to better consistence with ERAI compared to Control (without spectral nudging), especially for U and T (Figure 9a, 9b and 9c). But SNIowU and SNIowT have comparable results with Control, and the reason is referred to the reply to Comment 4. For specific humidity (Figure 9d), all simulations have smaller Q in the whole layer, indicating that WRF model can effectively reduce the wet bias of water vapor of ERAI over the TP.

In order to quantify how well the nudged variables matched the ERAI, statistical error metrics of column-averaged U, V, T and Q derived from seven WRF simulations versus ERAI over the TP for the study period are given in Table 2. Compared to Control, the use of spectral nudging technique obviously improved the consistence between regional model field and driving field, with lower RMSEs of the constrained variables.

Aforementioned new Figure 9, new Table 2 and relevant descriptions have been added in 'Section 4.3' (Line: 332-369) in the revised manuscript.

Figure 9: Vertical profile of difference fields between ERA-Interim and seven WRF simulations averaged over the study period for (a) zonal wind component (U; m s-1), (b) meridional wind component (V; m s-1), (c) atmospheric temperature (k), (d) specific humidity (kg kg-1) over the TP. The vertical gray solid line represents the value of 0.0 at each pressure layer.

Table 2: Statistical error metrics of column-averaged zonal wind component (U; m s-1), meridional wind component (V; m s-1), atmospheric temperature (k) and specific humidity (kg kg-1) derived from seven WRF simulations versus ERA-Interim (ERAI) over the TP for the study period.

|                          |      | Control | SN    | SNIowU | SNIowT | SNnoT | SNQ_trop25 | SNQ_trop39 |
|--------------------------|------|---------|-------|--------|--------|-------|------------|------------|
| U (m s -1 )   | MB   | 1.61    | 0.29  | 1.57   | 1.49   | 0.31  | 0.30       | 0.26       |
|                          | RMSE | 2.06    | 0.80  | 2.03   | 1.97   | 0.79  | 0.82       | 0.81       |
| V (m s -1 )   | MB   | 0.71    | 0.18  | 0.65   | 0.67   | 0.23  | 0.26       | 0.19       |
|                          | RMSE | 1.87    | 0.99  | 1.78   | 1.80   | 1.04  | 1.02       | 1.03       |
| T (k)                    | MB   | -0.35   | 0.01  | -0.33  | -0.31  | -0.02 | 0.22       | 0.03       |
|                          | RMSE | 0.78    | 0.24  | 0.76   | 0.74   | 0.50  | 0.44       | 0.24       |
| Q (kg kg -1 ) | MB   | -0.05   | -0.08 | -0.06  | -0.06  | -0.09 | -0.03      | -0.05      |
|                          | RMSE | 0.29    | 0.35  | 0.30   | 0.29   | 0.27  | 0.22       | 0.25       |

"2. How the simulations of air temperature and water vapor mixing ratio are changed near the "lid" layer where the nudging is restricted toward both variables?"

Author Response: Thank you for this comment. We compared mean atmospheric

temperature and specific humidity between model level of 25 and 39 derived from SNQtrop25 and SNQtrop39 to reveal the impact of limiting nudging of temperature and water vapor mixing ratio. Their differences (SNQ\_trop25 minus SNQ\_trop39) are displayed in Figure R1. In Figure R1a, restricting nudging of temperature generally leads to a higher atmospheric temperature, with an exception over the northern Tibetan Plateau. In terms of specific humidity, restricting nudging of water vapor mixing ratio has an overall larger specific humidity (Figure R1b). The evaluations indicate that simulation without nudging of temperature produces higher temperature, while simulation without nudging of water vapor mixing ratio produces wetter atmosphere.

Figure R1. Difference of (a) atmospheric temperature (K) and (b) specific humidity (kg kg-1) averaged over the study period derived from SNQ\_trop25 minus SNQ\_trop39.

"3. The large simulated precipitation bias and high precipitation RMSE over the Himalaya mountain region shown in Figure 2 and Figure 4 are possibly caused by the topographic differences between ERAI and WRF model. What are the impacts of nudging toward geopotential height on simulated results?"

**Author Response:** Thank you for this comment. Owing to the sharply lifted topography height and th

---

## Author Response (AR1)

Dear Editor,

On behalf of my co-authors, I'm submitting our revised manuscript for possible publication in "Geoscientific Model Development".

Thank you very much for your great efforts and high efficiency on evaluating our manuscript. According to Executive editor's suggestion, our paper title has been changed to "Development and evaluation of spectral nudging strategy for the simulation of summer precipitation over the Tibetan Plateau using WRF (v4.0)". We would also like to sincerely thank two anonymous reviewers for their constructive comments, which are very helpful for us to improve our manuscript. We have fully addressed each comment and revised the manuscript accordingly. In the following response, the referees' comments are listed in black, while our responses are marked in blue. A marked-up manuscript version showing the changes made has also been uploaded for review. We also mention where we make necessary changes in the revised manuscript by indicating line numbers in our responses. The line numbers correspond to the clean version of the revised manuscript.

We look forward to hearing from you soon.

Yours truly, Lei Zhong et al.

**Response to Anonymous Referee #1**

We would like to thank Referee #1 for the thorough review and constructive comments. We have carefully considered and addressed all comments, and significantly revised our manuscript. Please find our point-by-point response below. For clarity, the reviewer's comments are listed in **black**, while our response are shown in blue.

"A nicely written manuscript about downscaling of global reanalyses for regional climate modeling studies. I have no major comments, there are some spelling mistakes and incomplete sentences, but I am sure they will get corrected during the editing process."

**Author Response:** We would like to thank the reviewer for your positive comments on our study. We have taken each comment into account and revised our manuscript accordingly.

"I was missing a slightly more in-depth dynamical interpretation of why the best performing set of parameters improved the skill of WRF as an RCM."

**Author Response:** Thank you for this comment. To address this question, it is necessary to clarify the systematic inability of regional climate models first. Generally, a full forecast model within the regional domain can be decomposed into two components: the base field and the perturbation field. The base field is obtained from the driving fields, while the regional perturbation field is defined as the difference between the total field and the base field (Juang and Kanamitsu, 1994). Because of the systematic error of RCM, the inconsistencies between the model solution and the driving fields along the boundaries can cause large-scale errors (e.g., erroneous long waves), which might develop within the regional domain and produce undesirable noise.

In this case, spectral nudging technique is used to incorporate the large internal scales from outer model into the regional domain by adding a nudging term to the tendencies of nudged variables in model fields (Waldron et al., 1996). The nudging term is the summation

of the difference between the spectrally expanded base field and the model field, multiplied by the nudging coefficients over selected wavenumbers. By applying spectral nudging technique, perturbation tendencies (a difference between the computed full field tendencies and the base field tendencies) are transformed to spectral space in RCM. Then, the spectral truncation filters out those waves longer than the regional domain from the perturbation tendencies. As a result, any scales longer than the regional domain cannot be modified during the course of integration (Kanamaru and Kanamitsu, 2007). Therefore, the large-scale information is retained and small-scale information can be freely resolved in downscaling process.

From the dynamical perspective, the best performing set of parameters of spectral nudging in this study is a combination of three corrections. For horizontal winds, spectral nudging determines the tendency of wind field perturbation whose scale was larger than the wavelength of 1000 km, so that the inconsistencies between the regional field and the base field can be alleviated. Nudging to geopotential height corrects the erroneous pressure difference due to the surface elevation difference between the ERA-Interim and WRF model, leading to a better resolved topography of the Himalayas in the inner domain. For the temperature and humidity, their nudging coefficients were set to zero, which means their mean perturbations were set to zero, preventing the external forces on moisture and temperature (from nudging) to affect small-scale dynamics. In summary, nudging to winds and geopotential is strong enough to reduce large-scale errors in the regional domain, and this method improves the skill of WRF as a RCM.

Aforementioned dynamic interpretations of how spectral nudging influences the solution of RCM and how the set of spectral nudging parameters improved the skill of WRF have been added in 'Section 2.2' (Line 126-130), 'Section 4.3' (Line 332-334) and 'Section 5' (Line 384-388) in the revised manuscript, respectively.

Line 126-130: 'Spectral nudging, as a dynamical downscaling technique, is used to retain all the large-scale information from the driving fields, and to add smaller-scale information that the coarse driving fields can not resolve. By adding a nudging term on the variable of model field, perturbation tendencies (differences between the regional field tendency and

the driving field tendency) are dampened by spectral nudging at selected spatial scale, and then the inconsistence between model solution and the driving coarse field is alleviated.

Line 332-334: 'Therefore, the reason why not nudging towards temperature and moisture at all is preventing the external forces on moisture and temperature (from nudging), so that their small-scale dynamical features were not affected.'

Line 384-388: 'In the case of downscaling the coarse-resolution reanalysis, regional climate simulations suffer from the large-scale errors due to the inconsistencies between the model solution and the driving field along the boundaries because of the systematic error of RCM (Miguez-Macho et al., 2004;Waldron et al., 1996). The spectral nudging technique is thus used to relax the specific spectral scale of variables in model field towards the driving field, in order to alleviate the inconsistencies.'

"Line 117-118: (incomplete sentence) Why did you use Yonsei scheme? I am assuming it is for the PBL."

**Author Response:** Thank you for this comment. A misleading sentence exists in the original manuscript. Actually, most physical options used in this study are the same as the options of the High Asia Refined (HAR) data. But for PBL scheme, we selected the YSU PBL scheme rather than Eta similarity PBL scheme. In the revised manuscript, relevant content has been revised as follows:

Line 117-118: 'The Yonsei University scheme (Hong et al., 2006) was used for the PBL scheme.'

"Line 145: Is the 'ktrop' option for spectral nudging now available to the larger audience of WRF users?"

**Author Response:** Thank you for this comment. To our knowledge, this new 'ktrop' option has been added in the released WRF model v4.0, and WRF users can use this option via

'namelist.input'. More details and usages about 'ktrop' are available from the source code 'WRF/phys/module\_fdda\_spnudging.F' (https://github.com/wrfmodel/WRF/blob/master/phys/module\_fdda\_spnudging.F#L266).

"Line 276: 'transporting emissions - which emissions?"

Author Response: Thank you for this comment. The 'emissions' here means water vapor. In the revised manuscript, relevant text now reads:

Line 300-303: 'Deep convection favors the process of transporting water vapor into the upper-level atmosphere, through which moisture flux is vertically released into the atmosphere and influences the formation of precipitation during summer monsoon season (Fu et al., 2006; Heath and Fuelberg, 2014).'

"Line 285: By 'drag force', do you mean orographic blocking? What is the convergence of the water vapor? Please clarify."

**Author Response:** Yes, 'drag force' here is relevant to the orographic blocking. Specifically, it means the sub-grid orographic drag on water vapor due to the complex terrain over the southern slope of the Himalayas.

The convergence of water vapor is mainly caused by the barrier effect of the Himalayas, which are more than 5,000 m on average. In summer, abundant northward water vapor transport, originating from the surrounding oceans, impinges on the Himalayas. Upslope moisture transport is strictly limited by the sufficiently high orograph, and thus most of water vapor are 'wrung out' here, causing the convergence of water vapor (Dong et al., 2016). To our knowledge, model with coarse spatial resolution ignores the impact of mesoscale and microscale orography. Accordingly, the surface friction and turbulent form drag on airflow with water vapor are reduced (Wang et al., 2020). Therefore, the convergence and condensation of water vapor over the southern slope of the Himalayas are weakened. In this case, more water vapor can be transported to the higher area and even the interior of

the Tibetan Plateau.

Relevant text has been added to the revised manuscript as follows:

Line 265-274: 'The terrain of the Himalayas, featured by a sharp topography gradient, is more than 5,000 m on average and is regarded as a natural barrier for northward atmospheric flow. In summer, large amount of atmospheric water vapor transport, originating from the surrounding oceans, impinges on the Himalayas. The sufficient high topography strictly limits the upslope water vapor transport, and strong upward motions are consequently formed by the lifting effect of the complex terrain (Dong et al., 2016). However, the complex orography of the southern slope of the Himalayas is greatly smoothed in current RCMs. Accordingly, the surface friction and sub-grid orographic drag due to the impact of mesoscale and microscale orography on the airflow are weakened, which in turn reduce the convergence and condensation of water vapor over the southern slope of the Himalayas. Consequently, more water vapor transport could arrive the high-latitude TP, causing more precipitation over the TP.'

"Line 315: It would be very instructive to overlay the 'ktrop' model level on figures 7 and 8. Is WRF using a terrain-following vertical coordinate? I imagine that the 'lid' is not horizontal at all - does that change your conclusions about how and where the nudging is applied?"

**Author Response:** Thanks for your comments and suggestions. As shown in new Figure 7 and Figure 8, the 'ktrop' model level has been added in the revised manuscript and relevant explanation is available at Line 304-307.

Line 304-307: 'In order to clearly distinguish the impact of restricting nudging of temperature and moisture, black solid lines that representing the height of 'ktrop' layer of 25 (near the height of 9.6 km) and 39 (near the height of 16.1 km) were plotted in Figure 7 and Figure 8.'

In this study, the WRF model used "Hybrid Vertical Coordinate". As explained by the WRF

source code documentation 'README.hybrid\_vert\_coord' (https://github.com/NCAR/WRFV3/blob/master/README.hybrid\_vert\_coord), the hybrid vertical coordinate are terrain following near the surface, but then relax towards an isobaric surface aloft. The purpose of this coordinate option is to reduce the artificial influence of topography towards the top of the model.

Yes, you are correct. This lid is indeed not horizontal. Ideally, the user-defined lid (KTROP) is selected to represent the tropopause, and will be refined or supplemented with the space and time varying tropopause. The variability of the tropopause over the Tibetan Plateau (TP) is complicated during summer. In this study, it is an effective way to select a certain pressure layer in the model vertical layers representing the tropopause as the lid. It does not change our conclusions about how and where the nudging is applied.

"Line 395: If I understand you correctly, the best results were achieved with SNnoT, nudging only the winds and geopotential, from the top of the PBL through the domain top, while not forcing the temperature and humidity at all?"

**Author Response:** Thank you for this comment. Yes, your understanding are correct. The best results in improving the simulation of the intensity and diurnal cycle of precipitation were achieved when spectral nudging was applied towards winds and geopotential and not forcing the temperature and humidity at all. The reasons can be summarized as follows: In this study, we have already run the sensitive experiments of nudging toward temperature and humidity on precipitation simulations, such as:

EXP1, 'SN', with nudging towards horizontal wind components (U, V), temperature (T) and geopotential height (G), and their nudging coefficients were all set to 0.0003 s-1.

EXP2, 'SNIowT' with nudging towards U, V, T and G, but the nudging coefficient of T was 0.000045 s-1;

EXP3, 'SNnoT' with nudging toward U, V and G only;

EXP3, 'SNQ\_trop25' with nudging towards U, V, T, G and Q, while restricting nudging towards T and Q above the model layer of 25 (the lower limit for the 'lid' layer);

EXP4, 'SNQ\_trop39' is same as 'SNQ\_trop25' but restricting nudging towards T and Q above the model layer of 39 (approximate the tropopause layer).

A new Figure 6 has been given to evaluate above experiments for column-integrated water vapor transport. Compared to ERAI (Figure 6a1) and ERA5 (Figure 6b1), SN (Figure 6d1) misrepresented the large-scale northward water vapor transport. The impact of nudging towards temperature can be found in the comparison between SNIowT (Figure 6f1) and SNnoT (Figure 6g1), in which SNIowT simulated much stronger northward water vapor transport than SNnoT over the southeastern TP. This excessive water vapor transport can also be observed in SNQ\_trop25 (Figure 6h1 and Figure 6h2) and SNQ\_trop39 (Figure 6i1 and Figure 6i2).

A new Figure 7 has been given to evaluate the impact of nudging on convection over the southern slope of the Himalayas. From the Figure 7g, the strongest upward wind was simulated by SNQ\_trop39, followed by SN (Figure 7b) and SNQ\_trop25 (Figure 7f). In this case, large amounts of water vapor can be transported to the upper troposphere by strong upward motion, and then conveyed to the interior of the TP through upper-level atmospheric circulation. The upward motion over the southern slope of the Himalayas simulated by the SNnoT (Figure 7e) showed a clear reduction. Therefore, most of the water vapor were condensed in upslope flow over the Himalayas, causing less water vapor available for precipitation over the interior TP.

Accordingly, the smallest RMSE and MAE of precipitation simulation were achieved by SNnoT (Figure 4) rather than SNIowT, SNQ\_trop25 or SNQ\_trop39. In addition, larger RMSEs and MAEs were obtained when nudging towards humidity was applied. Therefore, spectral nudging was applied towards winds and geopotential only while not forcing the temperature and humidity at all.

Aforementioned new Figure 6 and Figure 7 have been added in the revised manuscript and relevant descriptions have also been added in 'Section 4.1' (Line 275-296) and 'Section 4.2' (Line 311-327), respectively.